# Anchor objects drive realism while diagnostic objects drive categorization in GAN generated scenes
Aylin Kallmayer ⬡ ✉ & Melissa L.-H. Võ ⬡

Our visual surroundings are highly complex. Despite this, we understand and navigate them effortlessly. This requires transforming incoming sensory information into representations that not only span low- to high-level visual features (e.g., edges, object parts, objects), but likely also reflect co-occurrence statistics of objects in real-world scenes. Here, so-called anchor objects are defined as being highly predictive of the location and identity of frequently co-occuring (usually smaller) objects, derived from object clustering statistics in real-world scenes, while so-called diagnostic objects are predictive of the larger semantic context (i.e., scene category). Across two studies ($N_1 = 50$, $N_2 = 44$), we investigate which of these properties underlie scene understanding across two dimensions – realism and categorisation – using scenes generated from Generative Adversarial Networks (GANs) which naturally vary along these dimensions. We show that anchor objects and mainly high-level features extracted from a range of pre-trained deep neural networks (DNNs) drove realism both at first glance and after initial processing. Categorisation performance was mainly determined by diagnostic objects, regardless of realism, at first glance and after initial processing. Our results are testament to the visual system's ability to pick up on reliable, category specific sources of information that are flexible towards disturbances across the visual feature-hierarchy.

Despite their complexity, humans are incredibly efficient at understanding natural scenes. From deriving global scene properties at first glance to guiding attention during visual search, information processing at every stage seems effortless[1–10]. A large body of research has identified many routes towards efficient scene processing, often considering the contribution of different sources of information across time.

Scene categorization, i.e., the processes of transforming retinal input into semantically rich categories, has long been considered a key capacity of the visual system[11,12]. It is a fast and automatic process, relying on the analysis of local information such as objects, abstract features like scene functions, as well as global summary statistics or gist[11,13–18]. In recent years, feature hierarchies – from low-level edges and oriented lines to high-level visual features like object parts and whole objects[19] (see Supplementary Fig. 2 for high-level visual feature visualizations) – have been quantified from activation patterns in deep neural network (DNN) layers. These feature spaces can be used to predict the spatiotemporal dynamics of the content and structure of neural representational spaces underlying visual processing[19–21].

While interactive object scene processing has long been considered a key component of the visual system[22–25], object-to-object relations have

recently gained more traction, as co-occurrence statistics in both language and vision have been found to be represented in core object representations of the ventral stream[26,27]. It is likely that such relations are crucial for scene processing as well[22], as they affect not only predictions about which objects can be expected in a scene, but importantly, predictions about their configurations. These relationships have recently been conceptualized into the framework of scene grammar[8]. Here, scenes are decomposed into clusters of frequently co-occurring objects, coined phrases. These conceptual units consist of so-called anchor objects (e.g., a sink), which predict the identity and location of other smaller objects within the phrase (e.g., a toothbrush). Anchor objects have been found to guide attention and locomotion through real-world scenes[28–30] and are characterized by four properties: (1) the frequency in which objects appear together, (2) the distance between objects, (3) the variance of the spatial location, and (4) clustering of objects within scenes[9,28].

Anchor and diagnostic object properties have previously been operationalized into scores: Diagnosticity represents the probability that an image belongs to a scene category given the presence of that object, anchor status frequency represents the probability with which an object has the status of being an anchor in a scene category[23,27].

Goethe University Frankfurt, Department of Psychology, Frankfurt am Main, Germany. ✉e-mail: kallmayer@psych.uni-frankfurt.de

Anchor objects can be diagnostic and the other way around, though the two differ in their main function: Diagnostic objects allow inferring the semantics of the scene as a whole while anchor objects – which are usually big and stationary – can be easily resolved in the visual periphery and thus can efficiently guide attention to smaller objects that we interact with during real-world search. Therefore, we will consider both as sources of information for the present study, disentangling individual and shared contributions for different aspects of scene understanding.

In the present study, we used images generated from generative adversarial networks (GANs)[31] (Fig. 1a) to probe the contribution of visual features and specific object types to scene understanding along two dimensions – realness and category specificity. GANs are a class of generative neural networks that learn to generate new samples from the distribution of training images e.g., natural indoor scenes. For this, they need to learn the core components and their composition that make a scene. GAN dissection[32] has demonstrated the emergence of generator units that code for specific objects (structural elements as well as diagnostic objects), providing evidence that GANs indeed do pick up core scene ingredients at the object level.

Generated images are inherently ambiguous and naturally vary in (at least) two dimensions important for scene understanding: First, they vary in how photorealistic they appear. Second, in the case of GANs trained on indoor scenes, they vary in their scene category specificity. The two are most probably correlated (e.g., it might be easier to categorize an image with fewer visual artefacts) but a generated indoor scene that looks photorealistic might still not be easily categorized. On the other hand, an obviously generated image that contains a lot of artefacts might still be clearly categorized as a kitchen scene. We make use of this naturally occurring variance in generated images that allows us to probe exactly what kind of information across the visual processing hierarchy is used to understand scenes, bringing together features extracted from a range of DNNs as well as specific object types representing real-world co-occurrence statistics, i.e., anchor status frequency and diagnosticity. What makes a scene real, what makes it categorizable, and how are these two connected? Are they solely dependent on the presence (or absence) of low- to high-level visual artefacts, like disturbances in texture and contours, or does the visual system rely on a certain object structure following real-world co-occurrence statistics?

Participants viewed real and generated images for 50 ms or 500 ms across two online experiments (Fig. 1b). We considered brief and long presentation durations to probe behavior at gist-level processing as well as at initial foveal sampling once the scene's gist has been extracted. We slightly increased the shorter presentation duration from what is usually considered to be needed to detect initial meaning[5] as we did not know how using generated images would affect these previously found thresholds. In Experiment 1, we operationalized realism via two different scores. First, participants performed a two-alternative forced choice task (2AFC) detecting real amongst generated images. Second, participants rated how realistic generated images appeared on a scale from 1 to 6 with no time constraints. From this, we modeled responses (1=real, 0=generated) and ratings from our features at different presentation durations. In Experiment 2, we let participants perform a 5-way alternative forced choice scene categorization task, this time categorization performance being the score of interest. We assumed that while both low- and high-level DNN features could explain realism and categorization performance to a certain degree, specific object types reflecting real-world regularities would be especially useful at resolving uncertainty.

## Methods
The studies presented were not preregistered.

### Participants
Fifty participants completed Experiment 1 (36 women, 14 men, 0 non-binary participants, 0 participants with undisclosed gender, M = 20.74 years old, SD = 2.5) and 44 participants completed Experiment 2 (30 women, 14 men, 0 non-binary participants, 0 participants with undisclosed gender,

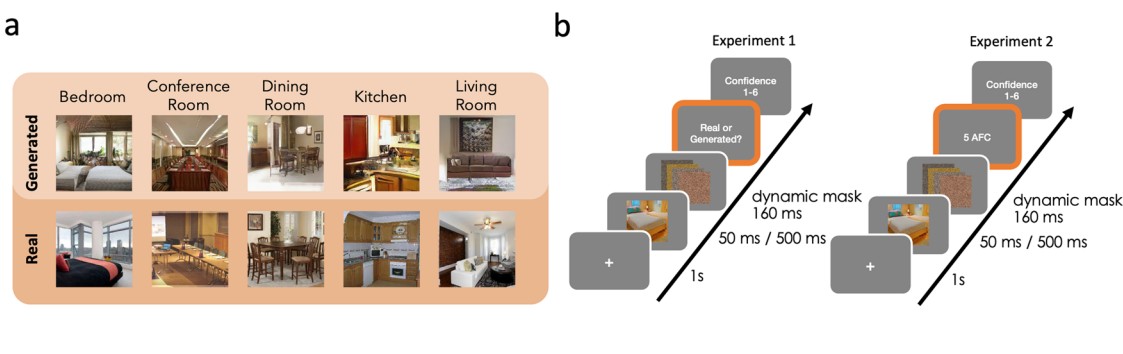

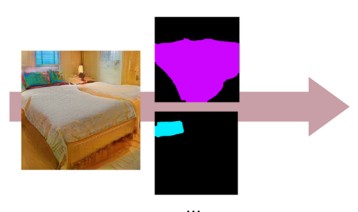

**Fig. 1 | Stimuli, trial sequences, and segmentation approach. a** Examples for real and generated images used in the present study. Generated images were generated from 5 different progressive generative adversarial networks (GANs)[32] each trained on one of the five respective LSUN scene categories[34]. Real images were randomly chosen from LSUN validation sets. The image set consisted of 30 real and 30 generated images from each category. **b** Trial sequences for part one of Experiment 1 (left) and Experiment 2 (right). Procedures differed only in terms of the task performed by participants, but all parameters related to stimuli presentation were kept the same. **c** Images were passed through a segmentation network[85] to obtain object predictions. For each image, all predicted objects were matched with an external database to assign anchor frequency and diagnosticity scores based on precomputed probabilities given the object and scene category. Each scene was then assigned the maximum score from all its predicted objects.

M = 23.2 years old, SD = 5.3). Age and gender were provided by participants via an online form, we did not collect any information on race/ethnicity. Prior power analyses suggested 50 participants for both experiments. Six participants had to be excluded from Experiment 2 because they aborted the experiment before completing all trials. Participants were recruited online via SONA and received course credit for participation. Normal or correct-to-normal vision was stated as condition to participate, however, participants did not have to perform any tests prior to participation. Participants were unfamiliar with the stimulus material and could only participate in either Experiment 1 or Experiment 2. Therefore, there were no participants that participated in both experiments. Informed consent was given via an online form before the experiments. All aspects of data collection and analysis were carried out in accordance with guidelines approved by the Human Research Ethics Committee at Goethe University Frankfurt.

## Stimuli and design
We collected 150 generated and 150 real photographic images of indoor scenes from five categories with 30 images per category (bedroom, conference room, dining room, kitchen, living room). We used progressive generative adversarial networks (PROGGANs)[33] pre-trained[32] on respective LSUN[34] categories to generate images for each category. Images were generated by randomly sampling from the latent spaces of the pretrained GANs. Code to generate the same set of images we used in this study can be found via the Open Science Forum (OSF) repository (see Data Availability section). We did not perform any further selection after generating from the random sample. Therefore, we did not remove or replace any of the sampled images, even if they contained artefacts. Real images were randomly selected from the LSUN validation image sets for each category. Images that depicted people, animals, or faces, as well as images containing watermarks or other form of added text were exchanged. Examples of images used in both experiments can be seen in Fig. 1a. All stimuli are available via the OSF repository (see Data Availability section). In Experiment 1, we used the full set of 150 generated and 150 real images, in Experiment 2, we included the full set of generated images and randomly sampled a subset of 50 real images for each participant (30 per category). In both experiments, we employed a dynamic masking paradigm consisting of four masks that were presented in rapid succession (40 ms each). Masks were created by randomly rearranging pixels of each real and generated image. Masks were then randomly assigned to trials for each participant. In both experiments, each image was presented only once per participant either for 50 ms or 500 ms counterbalanced between participants.

## Apparatus and online data collection
Participants' screen size was determined with the credit card method, whereby participants matched the size of a credit card on screen to a real credit card. Participants were instructed to look for a quiet, dimly lit location and to assume a viewing distance of approximately 60 cm resulting in visual angles of approximately 9.5° both horizontally and vertically for all stimuli (assuming a viewing distance of 60 cm). While variation in viewing distance and thus variation in visual angle cannot be ruled out, we expect variations to be minimal and if at all have similar effects on all conditions. The experiments were programmed using PsychoPy[35] (v2023.1.0) and hosted on Pavlovia (https://pavlovia.org).

## Procedure
In both experiments, each trial sequence (Fig. 1b) was initiated by a central fixation cross presented on screen for one second. Then, the image (real/generated) appeared for either 50 ms or 500 ms followed by a dynamic mask for 160 ms. In Experiment 1, participants were instructed to press different keys for generated or real scenes. In Experiment 2, participants performed a five alternative forced choice (5-AFC) scene classification task (bedroom, conference room, dining room, kitchen, living room) using numbers 1–5 on their keyboards. Participants completed six practice trials. In both experiments, each response was followed by a confidence rating (1 = "not confident at all", 6 = "very confident").

In part two of Experiment 1, participants gave each generated image a rating from 1 ("not realistic at all") to 6 ("very realistic") with no timeout.

## Scene segmentation, anchor status frequency, and diagnosticity
In order to assign anchor status frequency and diagnosticity scores to each scene, we needed to identify generated objects. For this, we used an automated approach (Fig. 1c) that did not require human labeling. First, we passed each image through a pre-trained scene segmentation network[36] yielding a vector of predicted objects and respective probabilities. From predicted objects with network probabilities > 0.3 we removed structural elements such as windows, walls, floor, and doors. For each predicted object we then assigned precomputed probabilities – anchor status frequency (which represents the probability of a given object functioning as an anchor object in a given scene) and diagnosticity (which represents the probability that an image belongs to a scene category given the presence of that object)[23,27]. These probabilities were calculated from a large labeled image dataset[37]. For each scene, we then assigned the maximum score from all its predicted objects. To assert that our approach led to sensible scores, we showed two independent raters each scene together with the object names that received highest anchor status frequency and diagnosticity scores and let raters indicate if and where in the scene they could identify these objects. The results matched our scene segmentation results.

## Data analysis
We processed all data in R[38] (v4.1.2.) and used Python[39] (v2.3.492) adapting code from DeepDive[40] to extract and subsequently map deep neural network (DNN) feature activation maps to behavior. We used a semantic segmentation demo network from the MIT scene parsing benchmark[36] to automatically detect objects in our scenes.

In R, we used the lme4 package[41] (v.1.1.34) to employ (generalized) linear mixed effects models ((G)LMMs) to test for effects of presentation duration (50 ms/500 ms), image condition (real/generated), anchor status frequency (range: 0–1), and diagnosticity (0–1) on realness (Experiment 1) and categorization performance (Experiment 2). We chose this methodology due to its potential advantages compared to Analysis of Variance (ANOVA), as it enables simultaneous estimation of variance both by participant and by stimulus[41–43]. To establish the random effects structure for each model, we followed a stepwise approach, beginning with a full model containing varying intercepts and slopes for all by-participant and by-stimulus factors in our design[44]. Then, we iteratively removed random slopes that did not significantly contribute to model goodness of fit, as determined by likelihood ratio tests[45]. This strategy helped us avoid over-parameterization and yielded models that align well with the observed data. To promote converging models, we z-transformed (rescaled and centered) all continuous predictors.

For the LMM, we report $\beta$ regression coefficients with the t statistic and p values calculated with the Satterthwaite's degrees of freedom method using the lmerTest package[46] (v.3.1.3). We inspected the normal probability plot and power coefficient for the continuous rating variable using the MASS[47] package and the Box-Cox procedure[48] to meet LMM assumptions. As a result, the dependent variable was not transformed. Additionally, we report squared eta $\eta_p^2$ and 95% confidence intervals using the effectsize package (v.0.8.3)[49]. For the GLMMs, we report $\beta$ regression coefficients along with their corresponding z statistic and Wald's confidence intervals. P values are derived from asymptotic Wald tests. Note, that $\beta$ regression coefficients act as a standardized effect size measure in the GLMM. For all models, we perform two-tailed significance testing using a 5% error criterion. We employed sum contrasts for presentation duration (50 ms/500 ms) and image type (real/generated), with slope coefficients indicating differences between factor levels, while the intercept represents the grand mean. All (generalized) linear mixed effects models were followed up by Bayesian regression analysis using the BayesFactor package (v.0.9.12)[50,51]. Bayes factors were computed for the full model and all possible sub-models (subsequently removing a single term at a time) to a null model using default mixture-of-variance priors[51–54] and Monte Carlo integration with 50,000

samples. The null model was a model with an additive model on the random factor (participant) plus intercept (grand mean). In cases where computing Bayes factors for all possible sub-models was not feasible, we selectively compared sub-models based on results from the GLMMs. Sub-models always retained the random participant factor. When comparing individual effects, we use subscripts to indicate the direction of the comparison: whether the Bayes factor is the evidence for a full model relative to the appropriate restriction (i.e., $B_{10}$), or the reverse (i.e., $B_{01}$). We report AIC and % error for all model comparisons corresponding to proportional error estimate on the Bayes factor.

If indicated, post-hoc comparisons were performed by obtaining estimated marginal means (EMMs) and computing linear trend analysis (for interactions between continuous and categorical predictors).

We report linear trends together with Wald's confidence intervals. We used the ggplot2 package (v.3.4.2)[55] for graphics and emmeans (v.1.8.7)[56] for post-hoc comparisons.

We were interested in performance differences for real and generated images across presentation durations and tasks as well as which features would contribute to explaining this performance. We considered feature maps obtained from a range of neural networks trained on computer vision tasks such as classification, self-supervised contrastive learning, and language-pretrained contrastive learning as well as object centric features reflecting real-world co-occurrence statistics (anchor status frequency and diagnosticity) as explanatory candidates towards our behavioral observations. In the following sections, we will go into detail on each individual analysis.

ROC curves and AUC. In Experiment 1, participants performed a 2AFC task, detecting real amongst generated images for brief (50 ms) and long (500 ms) presentation durations. According to signal detection theory (SDT)[57] correctly labeling real images as real was classified as a hit, while labeling generated images as real was classified as a false-alarm (FA). In SDT, signal present/absent responses are based on internal response probability curves for noise trials (where signal is absent) and signal plus noise trials (where signal is present). Responses are given based on a criterion that can lie anywhere along the internal response axis. To quantify the ability to discriminate between real and generated images we computed empirical receiver-operating characteristic (ROC) curves, which capture the hit rate to FA rate ratio for different criterions. ROCs for each participant were computed based on the confidence ratings collected after each trial. This allowed us to compute a series of hit and FA rates instead of a single point measure (for an in depth explanation of the approach see Brady et al.[58]). We then used the pROC package[59] to build and subsequently compare ROC curves for the 50 ms and 500 ms conditions using bootstrap tests ($N = 2000$) with the alternative hypothesis that the true difference in area under the curve (AUC) is not equal to 0.

Realness. We considered two behavioral measures for realness. First, we predicted signal present/absent (real/generated) responses in our 2AFC task from interaction terms between the true image condition (real/generated), presentation duration (50 ms/500 ms), anchor status frequency (range: 0–1), and diagnosticity (range: 0–1). In the GLMM, interaction terms with the true image condition reflect the effect of each predictor on the discriminability index d'. Our final random effects structure had by participant and by stimulus random intercepts as well as by participant random slopes for presentation duration, true image condition, and diagnosticity, and by stimulus random slopes for presentation duration.

Second, we predicted realness ratings (1 = highly unrealistic, 6 = photorealistic) that we collected for generated images from interaction terms between anchor status frequency and diagnosticity in a LMM treating realness as a continuous variable. In our final model, we had by participant and by stimulus random intercepts and random slopes for diagnosticity, as well as by participant random slope for anchor status frequency.

Categorization. We again applied GLMMs with interaction terms for image type (real/generated), presentation duration (50 ms/500 ms), anchor status frequency (range: 0–1), diagnosticity (range: 0–1), and realness

(range: 0–1) to predict categorization accuracy (1 = correct/0 = incorrect). Realness in this case refers to the average response an image received in Experiment 1 (1 = real, 0 = generated) separately for each presentation duration condition. We included all possible up to 4-way interactions but excluded the 5-way interaction as it made the model fail to converge and the effects difficult to interpret.

Our final random effects structure had by participant and by stimulus random intercepts and random slopes for the effect of presentation duration and a by participant random slope for the effect of image type.

DNN features. To investigate how much variance in the observed behavior could be explained from variance in underlying feature spaces we deployed a range of deep neural networks (DNNs) pretrained on canonical computer vision tasks. We chose this approach over deploying a single model to obtain features that reflect different training styles and dataset constraints. The models we used were: Alexnet[60] (image classification trained on imagenet), VGG19[61] (image classification trained on imagenet), Resnet50[62] (residual learning, image classification trained on imagenet), GoogLenet[63] (image classification trained on imagenet), Taskonomy scene classification network[64] (transfer learning, scene classification MIT Places), Resnet50 clip (contrastive language image pre-training, hybrid language-vision model)[65], Resnet50 SimCLR (self-supervised contrastive learning)[66]. We linearly decoded behavioral responses (realness, categorization performance) from the network activity via ridge (L2 regularized) regression. We closely followed an approach by Conwell et al.[67] using layer-wise feature-maps as predictors in leave-one-out cross-validated ridge regression where we predicted average scores for each image. After obtaining network activations we used sparse random projection (SRP)[68,69] to reduce feature map dimensionality. We then correlated predicted values with actual values to obtain scores for each feature-map. Scores were binned into slices of 10 (from 0, earliest, to 1, deepest layer), taking the average score over layers in each bin. Instead of testing scores against zero, we tested against scores obtained from randomly initialized versions of our pretrained networks. We do this to account for the amount of variance that randomly initialized neural networks are able to explain in visual processing without any previous training[70].

We performed permutations tests for the mean difference between trained and randomly initialized neural networks for each bin. Here, we compare the observed mean difference to the distribution of mean differences across 10.000 permutations where an observed empirical difference larger than 95.5% of the permutation distribution is treated as statistically significant. We report bootstrapped means and 95% confidence intervals for differences between trained and randomly initialized neural networks for each bin. To account for multiple comparisons, we performed false discovery rate correction across bins. Additionally, we perform paired Bayesian t-tests to compare trained with randomly initialized models for each bin. We use default priors ($r = 707$) to test the null hypothesis ($m = 0$) against an alternative hypothesis suggesting non zero effect sizes ($r = 0.707$).

### Reporting summary
Further information on research design is available in the Nature Portfolio Reporting Summary linked to this article.

## Results
We will present behavioral results on the ability to categorize and discriminate between real and generated scenes for brief (50 ms) and long (500 ms) presentation durations. For each behavioral measure, we will go into the different factors that contributed to making scenes more realistic and categorizable, respectively. We will consider the contribution of low-through high-level visual features quantified from a range of deep neural networks (DNNs) trained on canonical computer vision tasks (such as object and scene classification and language-vision pre-training), as well as object centric features representing real-world co-occurrence statistics (anchor status frequency and diagnosticity) obtained from a scene segmentation procedure (Fig. 1c).

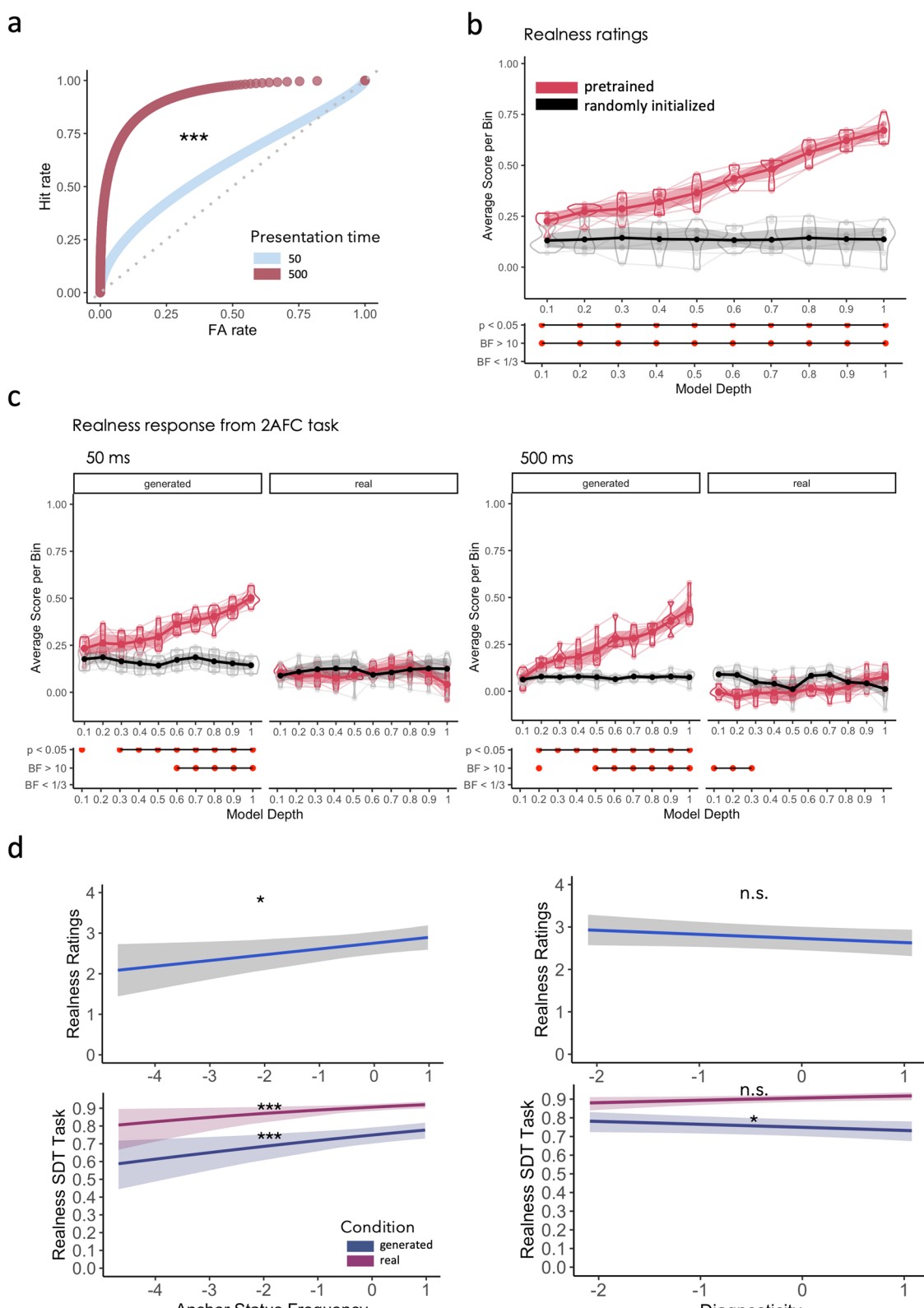

### Generated scenes appear real at first glance

In the 2AFC task, participants had to detect real amongst generated images for brief (50 ms) and long (500 ms) presentation durations. We modeled Receiver Operator Characteristics (ROC) curves which we obtained using confidence ratings as suggested by Brady et al.[58], where the area under the curve (AUC) or C statistic represents a more representative overall performance score for the binary classification task than accuracy as it takes into account performance at different criterions. The AUC score ranges in value from 0–1 where a score of 0 represents 100% misclassifications and a score of 1 represents only correct predictions. At 50 ms, participants performed only slightly above chance (AUC = 0.6) and became significantly better at the task in the 500 ms condition (AUC = 0.92, $p < 0.05$; Fig. 2a).

**Fig. 2 | Results Experiment 1. a** Receiver operator characteristics curves (ROC) for the 50 ms condition in light blue and 500 ms condition in red. Hits reflect correctly identified generated images while false alarms reflect real images that were classified as generated. **b** Predicting realness ratings from DNN features. We extracted layerwise feature-maps from a set of neural networks that were trained on canonical computer vision tasks such as object and scene classification. We then predicted realness ratings and responses in the 2AFC task from dimensionality reduced feature maps (using sparse random projection) in leave-one-out cross-validated ridge regression. We show the average scores (correlation between predicted and actual realness values) per bin (10 bins from 0, earliest, to 1, deepest layer). We compared pretrained networks (in red) to networks that received no training (randomly initialized weights, in black) which represent the lower bound. Shaded areas represent 95% bootstrapped confidence intervals ($N = 7$ pretrained models, $N = 7$ randomly initialized models). Bootstrapped means and confidence intervals were created by resampling 1000 times. We plot $p$ values and Bayes Factors for each bin (trained versus randomly initialized). **c** Predicting responses from the 2AFC task using the same method described above for the 50 ms condition and the 500 ms condition. **d** Partial effects plots for the main effect of anchor status frequency and diagnosticity on realness ratings and responses in the 2AFC task as well as the interaction between diagnosticity and image condition (real/generated) in the 2AFC task. Partial effects were obtained using the ggeffects package[86] ($N = 50$ participants).***$p < 0.001$, **$p < 0.01$, *$p < 0.05$.

That is, generated scenes appeared more realistic to participants at first glance but were easily discriminated from real scenes at longer presentation durations (also see Supplementary Fig. 1 for sensitivity and bias across presentation time).

### High-level visual features and anchor frequency explain realness

What made a scene appear real as opposed to generated to people? Regressing over features extracted from neural networks trained on a range of computer vision tasks explained a considerable amount of variance in task responses and ratings. In general, high-level features explained the most variance (Fig. 2b, see Supplementary Fig. 2 for example visualizations) when compared to untrained, randomly initialized instances with highest scores obtained for predicting realness ratings (maximum difference between trained and random: $\text{diff}_{bin10} = 0.53$, $p < 0.05$, $\text{CI}_{95\%} = [0.46, 0.6]$, $B_{10} = 3.3 \times 10^3$). High-level features predicted responses in the 2AFC task for generated images in both presentation duration conditions (50 ms max. $\text{diff}_{bin10} = 0.36$, $p < 0.05$, $\text{CI}_{95\%} = [0.32, 0.4]$, $B_{10} = 3.6 \times 10^3$; 500 ms max. $\text{diff}_{bin10} = 0.36$, $p < 0.05$, $\text{CI}_{95\%} = [0.3, 0.42]$, $B_{10} = 1.8 \times 10^3$) with inconclusive evidence for real images (50 ms max. $\text{diff}_{bin7} = 0.03$, $p = 0.56$, $\text{CI}_{95\%} = [-0.3, 0.08]$, $B_{10} = 0.3$; 500 ms max. $\text{diff}_{bin10} = 0.07$, $p = 0.07$, $\text{CI}_{95\%} = [0.004, 0.13]$, $B_{10} = 1$).

In the 2AFC task, anchor status frequency scores significantly contributed to making images appear more real (Fig. 2d), independent of image type, presentation time, and diagnosticity ($\beta = 0.18$, SE $= 0.06$, $z = 3.19$, $p = 0.001$, $\text{CI}_{95\%} = [.06, .29]$). As expected from the ROC curves, there was a significant interaction between presentation duration and true image condition which in the context of signal detection theory represents a significant increase in discriminability d' (d' is an estimate of signal strength and reflects both the separation and spread parameters of the noise and signal plus noise curves in a signal detection paradigm) with longer presentation duration ($\beta = 0.65$, SE $= 0.04$, $z = 18.01$, $p < 0.001$, $\text{CI}_{95\%} = [0.58, 0.72]$).

There was also a significant interaction between image type and diagnosticity ($\beta = -0.11$, SE $= 0.05$, $z = -2.25$, $p = 0.02$, $\text{CI}_{95\%} = [-0.22, -0.05]$; see Fig. 2d). That is, generated images with high diagnosticity were less likely to produce false alarms ($\text{trend}_{diagnosticity} = -0.15$, $\text{CI}_{95\%} = [-0.28, -0.02]$), than real images with high diagnosticity ($\text{trend}_{diagnosticity} = 0.08$, $\text{CI}_{95\%} = [-0.07, 0.24]$). Bayes factor analysis provided corroborating evidence: A model $M_1$ with main factors for true image condition, presentation duration, and anchor status frequency as well as interactions between true image condition and presentation duration and true image condition and diagnosticity was the most preferable one considering all sub-models compared to the null model $M_0$ ($B_{10} = 1.75 \times 10^{1340}$, $\text{AIC}_1 = 13395$, $\text{AIC}_0 = 13696$, %error $= 1.76$). Comparing this model with the full model $M_f$ that additionally includes a main effect for diagnosticity suggests evidence for a lack of the diagnosticity main effect ($B_{1f} = 4$, $\text{AIC}_f = 13398$, %error $= 2.41$; see also Supplementary Table 1).

When we modeled realness ratings for generated images from anchor status frequency and diagnosticity we found similar response patterns as in the 2AFC task. Anchor status frequency significantly predicted ratings ($\beta = 0.15$, SE $= 0.07$, $t = 2.25$, $p = 0.03$, $\eta_p^2 = 0.03$, $\text{CI}_{95\%} = [0.02, 0.28]$) while diagnosticity did not turn out to be significant ($\beta = -0.09$, SE $= 0.06$, $t = -1.5$, $p = 0.13$, $\eta_p^2 = 0.02$, $\text{CI}_{95\%} = [-0.21, 0.02]$). Subsequent Bayesian

factor analysis suggested that a full model $M_f$ including main effects for anchor status frequency and diagnosticity plus an interaction term was the most preferable one considering all sub-models compared to the null model $M_0$ ($B_{f0} = 1.65 \times 10^{564}$, $\text{AIC}_f = 25061$, $\text{AIC}_0 = 25061$, %error $= 0.21$). Comparing an anchor status frequency only model $M_1$ with a diagnosticity only model $M_2$ we find more evidence for the anchor status frequency model ($B_{12} = 1 \times 10^5$, $\text{AIC}_1 = 25060$, $\text{AIC}_2 = 25062$, %error $= 0.92$; see also Supplementary Table 2).

To summarize, discriminating between real and generated images seems to be mostly a high-level process that relies on differences in high-level visual features. Crucially anchor objects, but not diagnostic objects, seem to contribute to making a scene feel real across presentation durations and image type. Both anchor status frequency and diagnosticity effected realness ratings, with evidence pointing to a strong contribution of anchor status frequency compared to diagnosticity.

### High-level visual features and diagnosticity explain categorization performance

Mostly high-level features explained variance in scene categorization accuracy (compared to untrained, randomly initialized instances) for generated and real images in the 50 ms condition (maximum difference between trained and random: generated max. $\text{diff}_{bin10} = 0.18$, $p < 0.05$, $\text{CI}_{95\%} = [0.1, 0.26]$, $B_{10} = 11.16$; real max. $\text{diff}_9 = 0.1$, $p < 0.05$, $\text{CI}_{95\%} = [0.03, 1.7]$, $B_{10} = 5.16$) and for generated images in the 500 ms condition (max. $\text{diff}_{bin10} = 0.19$, $p < 0.05$, $\text{CI}_{95\%} = [0.1, 0.29]$, $B_{10} = 11.59$). Evidence was inconclusive for real images in the 500 ms condition (max. $\text{diff}_{bin4} = 0.03$, $p = 0.05$, $\text{CI}_{95\%} = [-0.02, 0.07]$, $B_{10} = 1.1$; Fig. 3a). These scores were considerably lower than when we modeled realness in Experiment 1. What made images easy or difficult to categorize additionally to the distribution of high-level visual features?

We found a main effect for realness as continuous predictor ($\beta = 0.48$, SE $= 0.16$, $z = 2.9$, $p = 0.004$, $\text{CI}_{95\%} = [0.16, 0.81]$; Fig. 3b), but not image condition (real/generated) ($\beta = 0.09$, SE $= 0.17$, $z = 0.57$, $p = 0.57$, $\text{CI}_{95\%} = [-0.24, 0.44]$). That is, images with higher realness scores were categorized more easily (Fig. 3b). The GLMM also yielded a significant main effect for presentation duration, with performance increasing in the 500 ms condition ($\beta = -1.07$, SE $= 0.16$, $z = -6.85$, $p < 0.001$, $\text{CI}_{95\%} = [-1.37, -0.76]$; Fig. 3b). Crucially, diagnosticity significantly predicted categorization accuracy across realness and presentation durations ($\beta = 0.53$, SE $= 0.16$, $z = 3.26$, $p = 0.001$, $\text{CI}_{95\%} = [0.21, 0.85]$; Fig. 3b). Therefore, categorization performance was generally better for more realistic images, was explained mostly from high-level features, and scaled with the amount of diagnostic object information while anchor objects had no effect ($\beta = 0.01$, SE $= 0.26$, $z = 0.05$, $p = 0.96$, $\text{CI}_{95\%} = [-0.49, 0.52]$). Subsequent Bayes factor analysis suggests that a full model $M_f$ with main factors for true image condition, presentation duration, realness scores, diagnosticity and anchor status frequency is the most preferable one considering all sub models compared to a null model $M_0$ ($B_{f0} = 1.26 \times 10^{263}$; $\text{AIC}_f = 7983$, $\text{AIC}_0 = 8072$, %error $= 0.74$). Comparing a model without diagnosticity $M_1$ with a model without anchor status frequency $M_2$ provides stronger evidence for the effect of diagnosticity ($B_{21} = 4.33 \times 10^{22}$, $\text{AIC}_1 = 7983$, $\text{AIC}_2 = 7971$, %error $= 1.42$; see also Supplementary Table 3).

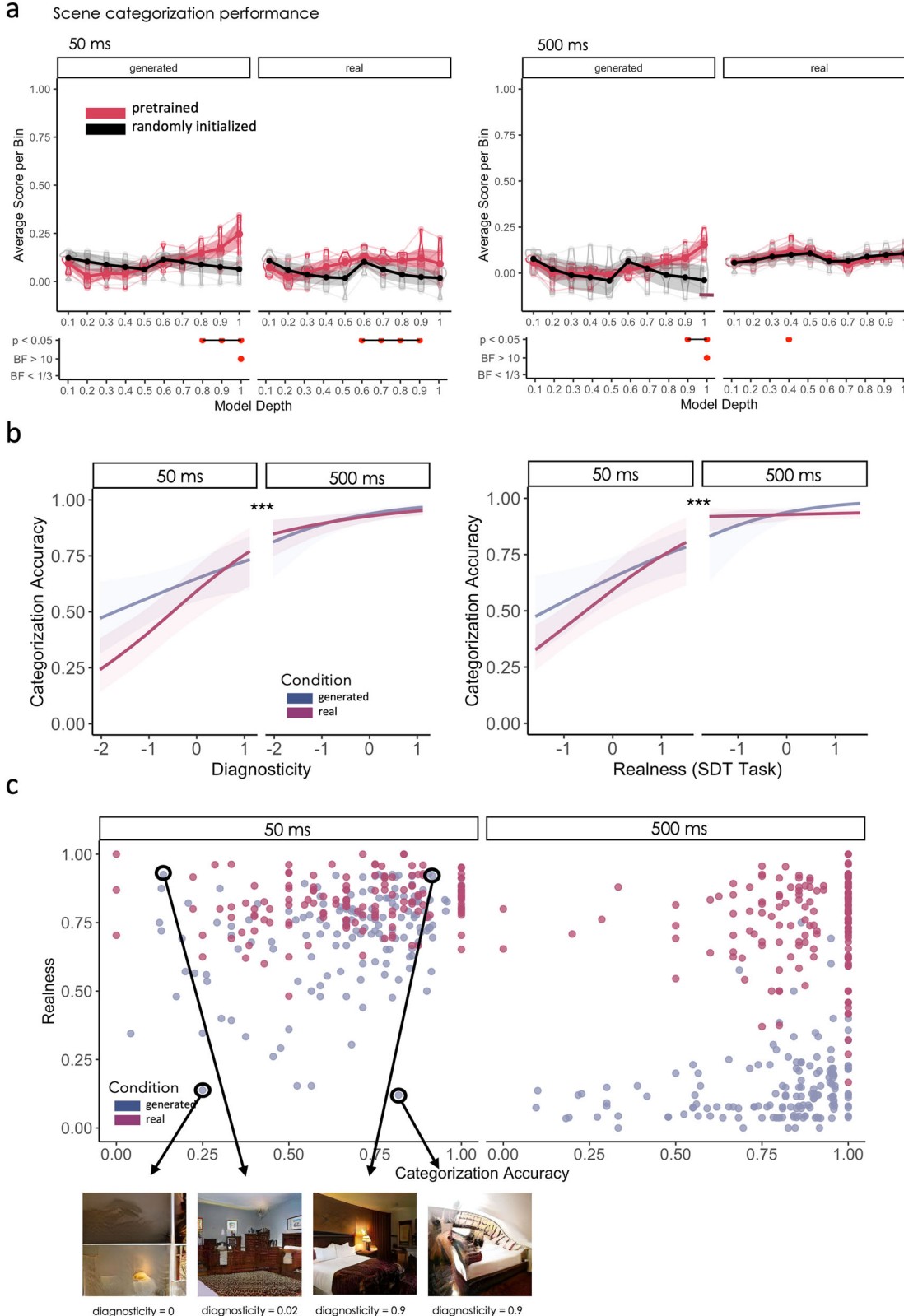

**Discussion**

In this study, we presented human observers with photographic scenes and scenes generated from Generative Adversarial Networks (GANs)[31] to learn about the contribution of different types of information towards quick and efficient natural scene understanding across two dimensions: realness and categorization. While mid- and high-level visual features extracted from

deep neural networks (DNNs) and specifically the presence of anchor objects contributed to making a scene real, diagnostic objects mainly contributed to increasing the scene's category specificity.

People are able to grasp a scene's gist (e.g., its basic level category, affordances, and global properties such as navigability), after a few milliseconds[1,17,18,71]. This fast extraction of meaning relies on both the feed-

**Fig. 3 | Results Experiment 2. a** We predicted categorization performance for each image from dimensionality reduced feature maps extracted from a range of deep neural networks via cross-validated ridge-regression. We show the average scores (correlation between predicted and actual realness values) per bin (10 bins from 0, earliest, to 1, deepest layer). We compared pretrained networks (in red) to networks that received no training (randomly initialized weights, in black) which represent the lower bound. Shaded areas represent 95% bootstrapped confidence intervals ($N = 7$ pretrained models, $N = 7$ randomly initialized models). Bootstrapped means and confidence intervals were created by resampling 1000 times. We plot $p$ values and Bayes Factor for each bin (trained versus randomly initialized). **b** Partial effects plots for the main effect of diagnosticity, presentation duration, and realness on categorization performance (**c**) Relationship between categorization performance, realness, and diagnosticity with examples for generated bedroom images with low realness and low categorization performance, high realness but low categorization performance, high realness and high categorization performance, low realness and high categorization performance, and corresponding diagnosticity scores. Partial effects were obtained using the ggeffects package[86] ($N = 44$ participants). ***$p < 0.001$, **$p < 0.01$, *$p < 0.05$.

forward processing of global scene statistics (e.g., statistical spatial layout information)[17,18] as well as the identification of objects and object constellations in the scene[11,16,22]. Both processes are assumed to interact with and constrain each other to support analysis at multiple processing levels[13,72]. Our study builds on previous studies on interactive object-scene processing by using ambiguous, generated scenes (that contain all of the "ingredients" of real scenes but are inherently less detailed and not always match expectations about reality) and consider realness and categorization as two separate, but related, dimensions of scene understanding.

After short presentation times of 50 ms, observers were not able to tell apart generated from real scenes. Here, anchor objects – large, stationary objects that are predictive of the location and identity of smaller surrounding objects – contributed to making a scene "feel" like a real scene. Unlike diagnostic objects – which can also be quite small (e.g., toothbrush in bathroom) – anchor objects tend to take up a larger proportion of the scene[22] and therefore contribute to its spatial layout (e.g., a cabinet in the kitchen). We argue that anchor objects inherently influence the statistical spatial layout information of a scene (without needing to be recognized) due to their size and structural properties[18,73] which in turn provide the basis for scrutinizing a scene's authenticity during swift feed-forward processing. We can assume that in the 50 ms presentation time condition backward masking largely prohibited recurrent processing and identification of individual objects in our already ambiguous scenes[74,75]. This was further supported by our computational modeling results: the feature hierarchy in DNNs captures increasingly abstract and discriminative features, from edges to textures and whole objects and their spatial arrangements, which all play into the global structure of the scene. We were able to explain up to 60% of variance in realness judgements from just high-level features (related to objects and their configurations, Supplementary Fig. 2). Later, generated scenes which seemed real after initial processing could be more easily discriminated from real scenes based on further recurrent analysis of high-level features and anchor objects (or lack thereof) which informed higher processing areas, in turn influencing downstream predictions and analysis at lower levels.

The presence of diagnostic objects, on the other hand, only slightly influenced how real scenes appeared, and interestingly did so in the opposite direction. This might seem counter-intuitive at first, but it really supports the idea that category specific information – which is what diagnosticity represents – can be abstracted away from any expectations regarding what the rest of the scene should look like and therefore poses a fast route towards categorization[22]. The strong effect of diagnostic objects, independent of realness, on categorization performance further supports this point: diagnostic objects supported fast scene categorization even if the global scene information (operationalized by the distribution of low- to high-level visual features) was disturbed and didn't match expectations about reality. It is a demonstration of the visual systems ability to pick up on latent factors in real-world scenes (object-scene co-occurrence statistics) which are processed at first glance and are reliable across situations of heightened uncertainty[11,16,22]. We found high-level visual features (Supplementary Fig. 2) only to be weakly predictive of categorization performance, independent of training (supervised, self-supervised, language-supervised) or dataset (imagenet, MIT scenes, 400 million image-text pairs). While diagnostic object-scene relationships do seem to be represented in DNNs trained on scene classification (and generation)[32,76] these relationships might not be

sufficiently disentangled in complex, high-level representations of deep DNN layers to predict fast categorization by the visual system. One might need to explicitly include more object-centric processing in computer vision models to achieve this[77,78]. On the other hand, our study might have lacked sufficient number of samples to learn a mapping from DNN features to behavioral scores for Experiment 2.

## Limitations
We intentionally used GANs that generated ambiguous images[32,33] instead of relying on state-of-the-art generative models which produce much more realistic images[79]. We are interested in finding a sweet spot where images are mostly recognizable but contain enough variance in the dimensions we are investigating (e.g., scene category specific information) so that we can experimentally test/probe contributing factors. Using a single GAN that is trained on multiple scene categories simultaneously could provide even more possibilities to investigate the types of information that allow to draw boundaries between representational categories[80].

Training a DNN on a deepfake detection task[81] and then applying interpretability tools, such as gradient visualization[82], to learn about which parts of the images bias deepfake detection presents an alternative way of quantifying features that distinguish real from generated images. One could enhance deepfake detection learning by comparing these biases to those identified in our current study on human participants.

## Conclusions
To conclude, anchor and diagnostic objects seem to contribute to scene understanding in different ways, that is, anchor objects may contribute to the distribution of low- to high-level visual features that make an authentic scene, while diagnostic objects allow fast and accurate categorization even in the face of hightened ambiguity due to noise in the image. Experimentally examining GAN generated images in vision studies provides a rich testbed which we can use to probe the emergence of structured scene representations. We believe that using GANs to generate and modulate images and then run them by the most powerful perception engine – our human observers – holds great potential to contribute to a better understanding of visual cognition in the real world. Importantly, using DNNs to learn about representations and computations in the human visual system will require testing of specific hypotheses in the context of experiments rather than pushing benchmarks for observational data[83,84].

## Data availability
All stimuli, experimental files, and raw data are available via Open Science Forum (OSF) under https://osf.io/x2rbq/?view_only=fbdb72f4a8904f9da e6d39d3e02f7cb5.

## Code availability
All analysis scripts, code to generate stimuli used in the present study, and PsychoPy files created to run the experiments online are available via the same OSF repository. https://osf.io/x2rbq/?view_only=fbdb72f4a8904f9da e6d39d3e02f7cb5.

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

## Acknowledgements
This work was funded by the Deutsche Forschungsgemeinschaft (DFG, German Research Foundation)—project number 222641018—SFB/TRR 135, sub-project C7 to M.L.V. and the Hessisches Ministerium für Wissenschaft und Kunst (HMWK; project 'The Adaptive Mind') and the Polytechnische Gesellschaft, Main Campus Doctus stipend awarded to A.K. The funders had no role in study design, data collection and analysis, decision to publish or preparation of the manuscript.

## Author contributions
Aylin Kallmayer: conceptualization, formal analysis, investigation, methodology, software, and writing – original draft and review & editing. Melissa L.-H. Võ: conceptualization, supervision, methodology, writing - review & editing.

## Funding

## Competing interests
The authors declare no competing interests.

 **10**
