## [Peer Review File · Communications Psychology]

19th Mar 24

Dear Ms Kallmayer,

Thank you for your patience during the peer-review process. Your manuscript titled "Making a scene – using GAN generated scenes to test the role of real-world co-occurrence statistics and hierarchical feature spaces in scene understanding." has now been seen by 2 reviewers, and I include their comments at the end of this message. They find your work of interest but raised some important points. We are interested in the possibility of publishing your study in *Communications Psychology*, but would like to consider your responses to these concerns and assess a revised manuscript before we make a final decision on publication.

We therefore invite you to revise and resubmit your manuscript, along with a point-by-point response to the reviewers. Please highlight all changes in the manuscript text file.

In addition to the referees' concerns, please ensure that your manuscript is compliant with our requirements for statistics reporting and interpretation, in particular concerning null results. You will find more information on these in the attached Editorial Request Table.

The Editorial Requests Table details critical reporting requirements for the revised manuscript. Please attend to each item and ensure your manuscript is fully compliant. We are requesting that your manuscript aligns with these requirements as this facilitates the evaluation of your manuscript, reducing delays in re-review and potential future acceptance. If your revised manuscript is not aligned with these requests on major issues, such as those concerning statistics, it may be returned to you for further revisions without re-review. Additional information can be found in our style and formatting guide *Communications Psychology* formatting guide.

Please use the following link to submit your

- revised manuscript,
- point-by-point response to the referees' comments,
- cover letter (as a separate document),
- the Editorial Policy Checklist (see below),
- the Reporting Summary (see below), and

- the completed Editorial Request Table (attached):

[link redacted]

Best regards,

Marike Schiffer

Marike Schiffer, PhD

Chief Editor

Communications Psychology

REVIEWER EXPERTISE:

Reviewer #1

Reviewer #2

REVIEWER REPORTS:

Reviewer #2 (Remarks to the Author):

The manuscript by Kallmayer and Vö presents a study conducted to elucidate factors affecting the perceived realness and categorisation of real-world scenes. Specifically, in line with the understanding of the relations among objects in scenes as a “scene grammar”, the authors focused on the role of anchor objects and diagnostic objects. The study involved using advanced computational methods for creating and analyzing the stimuli content, as well as conducting and analyzing behavioral data from two online experiments.

Overall, I think this is a well designed and conducted study, based on the authors previous work. While, in my opinion, it does not provide completely novel findings, I think the obtained results merit publication. I have only relatively minor comments and suggestions.

1. While I understand how the anchor and diagnostic objects were defined and how their impact on perception was estimated, it is not clear to me how the “high-level visual features’ extracted from DNNs can be defined. I would like to explicitly state that I am not very familiar with these methods and have never used DNNs myself, and thus I am wondering to what extent these features can be more precisely defined.

2. Method: “We modeled Receiver Operator Characteristics (ROC) curves which we obtained using confidence ratings as suggested by Brady et al., 29 where the area under the curve (AUC) or C statistic represents a more representative overall performance score for the binary classification task than accuracy as it takes into account performance at different criterions.”

While I understand that the proposed analysis might be favorable over accuracy scores, can you please elaborate what are the advantages of this method over an SDT analysis, in which perceptual sensitivity (d') and criterion (c) are calculated?

3. While in this study computational methods to address a vision science problem, the discussion seems strongly tilted towards more technical aspects. Perhaps the authors could include at least

one paragraph discussing their findings in a more “perceptual” context - i.e. how are these results relevant to our understanding of perception of scenes, gists, objects, and their interactions.

Minor issues:

Abstract - should be “underlie”

The authors refer to the discrimination task from exp. 1 as a “signal detection task”, but this is very unspecific as, to my understanding, any detection/discrimination can be called an SDT task. Using this term was quite confusing for me when I was reading the paper for the first time.

Fig. 1 - “Procedures only 126 differed in the task.” - not clear what this means, better “differed only in terms of the task performed by participants, but all parameters related to stimuli presentation were the same”, or sth similar

“there was a significant 168 interaction between presentation duration and true condition” - “true condition” is not clear in this context

Reviewer #3 (Remarks to the Author):

The authors investigated if humans can detect generated images from real images at short and long presentation times. They found generated scenes appeared realistic to participants at first glance but were easily discriminated from real scenes at longer presentation durations. They further showed that discriminating between real and generated images seems to be mostly a high level process.

Were the same images used in experiment 1 and 2? Were there any individuals who attended both experiments? If the images were the same, was the order of experiment 1 and 2 counter balanced across shared participants?

I think the procedure of stimulus selection should become more clear. For example, how would the authors generate an image of a specific category with a GAN and be happy with it? Not all the images generated by a GAN are very realistic, so I imagine multiple attempts have been made and then with some criteria the generated images were selected. Can the authors explain their approach?

What the authors think about the relation of presentation time, neural feedforward/feedback processing and detection of realistic images. Previous work has shown that briefly presented images and backward masking largely disrupt recurrent/feedback processing. I think this can be discussed in the discussion.

One interesting analysis would be to train a DNN to discriminate between real and fake scene images, and then use interpretability techniques to show what features were used by the DNN for this discrimination task (similar to Exp. 1).

EDITORIAL POLICIES

We ask that you ensure your manuscript complies with our editorial policies and reporting requirements.

To that end, we require revised manuscripts to be accompanied by two completed items: a reporting summary that collects information on study design and procedure, and an editorial policy checklist that verifies compliance with all required editorial policies.

Nature Research Reporting Summary

Editorial Policy Checklist

All points on the policy checklist must be addressed. Your revised manuscript can only be sent back to the referees if these checklists are completed and uploaded with the revision.

Notes: If you have submitted a Stage 1 Registered Report, Review, Primer, Comment, or Perspective you do not need to submit these forms. If you have already submitted these forms, you may disregard this request.

Reviewer #1:

The manuscript by Kallmayer and Vö presents a study conducted to elucidate factors affecting the perceived realness and categorisation of real-world scenes. Specifically, in line with the understanding of the relations among objects in scenes as a “scene grammar”, the authors focused on the role of anchor objects and diagnostic objects. The study involved using advanced computational methods for creating and analyzing the stimuli content, as well as conducting and analyzing behavioral data from two online experiments.

Overall, I think this is a well designed and conducted study, based on the authors previous work. While, in my opinion, it does not provide completely novel findings, I think the obtained results merit publication. I have only relatively minor comments and suggestions.

RE: Thanks a lot for the very positive feedback. We are very happy to hear that you see our work fit for publication at Communication Psychology.

1. While I understand how the anchor and diagnostic objects were defined and how their impact on perception was estimated, it is not clear to me how the “high-level visual features’ extracted from DNNs can be defined. I would like to explicitly state that I am not very familiar with these methods and have never used DNNs myself, and thus I am wondering to what extent these features can be more precisely defined.

RE: We thank the reviewer for their thoughtful comments and suggestions. High-level visual features in deep neural networks (DNNs) trained on scene classification can represent object parts, whole objects and configurations, as well as spatial layout. Nonetheless, these abstract representations are difficult to define in human interpretable terms. Low-level features usually reflect edges and orientations while high-level features combine features at lower levels to create abstract, semantic representations that are relevant for categorization. To help interpretation, there are a few visualisation techniques one can apply to trained convolutional DNNs. Therefore, we added supplementary figure 2 where we showcase results from two such visualisation techniques: Grad-Cam (Selvaraju et al., 2020) and class specific image generation (Simonyan et al., 2014; Yosinski et al., 2015). Grad-Cam produces ‘visual explanations’ for DNN predictions. Given some image and a pretrained convolutional DNN Grad-Cam returns a coarse localisation map for important regions in the image regarding one of the trained concepts. Class specific image generation is a set of methods implementing regularized optimization in image space, that is, generating an image which maximises the score for one particular class (e.g., creating an image which a DNN maximally classifies as “kitchen”). Both techniques allow interpretation of convolutional DNN features based on visualisations, but of course the full scope and complexity of such high-dimensional features remains quite elusive. Important for the present study is that realness prediction could be predicted quite well from high-level features while categorization performance seemed largely independent of these DNN features and more strongly related to diagnostic objects:

p.3 line 13: *“i.e., from low-level edges and oriented lines to high-level visual features including object parts and whole objects (Güçlü & Gerven, 2015) (see supplementary figure 2 for high-level visual feature visualizations)”*

We added a section where we elaborate on which aspects of high-level features contributed to realness and categorization performance in the discussion:

p.11 line 28: *“We argue that anchor objects inherently influence the statistical spatial layout information of a scene (without needing to be recognized) due to their size and structural properties^{20,38} which in turn provides the basis for scrutinizing a scene’s authenticity. We can assume that in the 50 ms presentation time condition backward masking largely prohibited recurrent processing and identification of individual objects in our already ambiguous scenes^{39,40}. This was further supported by our computational modelling results: the feature hierarchy in DNNs captures increasingly abstract and discriminative features, from edges to textures and whole objects and their spatial arrangements, which all play into the global structure of the scene. We were able to explain up to 60% of variance in realness judgements from just high-level features (related to objects and their configurations, supplementary Figure 2).”*

p.12 line 9: *“We found high-level visual features (supplementary Figure 2) only to be weakly predictive of categorization performance, independent of training (supervised, self-supervised, language-supervised) or dataset (imagenet, MIT scenes, 400 million image-text pairs). This fast, object-centric route to categorization therefore does not seem to be reflected in these DNN features. While diagnostic object-scene relationships do seem to be represented in DNNs trained on scene classification (and generation)^{32,41} these relationships might not be represented sufficiently disentangled in these complex, high-level representations of deep DNN layers the same way as they are in the visual system to support fast categorization.”*

2. Method: *“We modeled Receiver Operator Characteristics (ROC) curves which we obtained using confidence ratings as suggested by Brady et al., 29 where the area under the curve (AUC) or C statistic represents a more representative overall performance score for the binary classification task than accuracy as it takes into account performance at different criterions.”*

While I understand that the proposed analysis might be favorable over accuracy scores, can you please elaborate what are the advantages of this method over an SDT analysis, in which perceptual sensitivity (d') and criterion (c) are calculated?

RE: *We chose to compute AUC as a measure of accuracy across criterions, but added supplementary Figure 1 where we additionally display sensitivity and bias across presentation time conditions. For the main analysis we modelled sensitivity using mixed effects models (In the GLMM, interaction terms with the true image condition reflect the effect of each predictor on the discriminability index d' , p.15, line 41; As expected from the ROC curves, there was a significant interaction between presentation duration and true image condition which in the context of signal detection theory represents a significant increase in discriminability d' (d' is an estimate of signal strength and reflects both the separation and spread parameters of the noise and signal plus noise curves in a signal detection paradigm) with longer presentation duration ($\beta = .65$, $SE = .04$, $z = 18.01$, $p < 0.001$), p.6, line 36).*

We hope we clarified our analysis approach and how it is related to the computation of sensitivity (d').

3. While in this study computational methods to address a vision science problem, the discussion seems strongly tilted towards more technical aspects. Perhaps the authors could include at least one paragraph discussing their findings in a more “perceptual” context - i.e. how are these results relevant to our understanding of perception of scenes, gists, objects, and their interactions.

RE: We thank the reviewer for this important suggestion. Indeed, the discussion lacked a more nuanced consideration of our data from an object/scene perceptual processing perspective. We added the following paragraphs to the discussion and hope that we now consider these aspects more adequately.

p.11 line 12: *“People are able to grasp a scene’s gist (e.g., its basic level category, affordances, and global properties such as navigability), after a few milliseconds^{1,19,20,36}. This fast extraction of meaning relies on both the feed-forward processing of global scene statistics (e.g., statistical spatial layout information)^{19,20} as well as the identification of objects and object constellations in the scene^{10,16,18}. Both processes are assumed to interact with and constrain each other to support analysis at multiple processing levels^{12,37}. Our study goes beyond previous studies on interactive object-scene processing as we use ambiguous, generated scenes (that contain all of the “ingredients” of real scenes but are inherently less detailed and not always match expectations about reality) and consider realness and categorization as two separate, but related, dimensions of scene understanding.”*

p.11 line 28: *“We argue that anchor objects inherently influence the statistical spatial layout information of a scene (without needing to be recognized) due to their size and structural properties^{20,38} which in turn provide the basis for scrutinizing a scene’s authenticity. We can assume that in the 50 ms presentation time condition backward masking largely prohibited recurrent processing and identification of individual objects in our already ambiguous scenes^{39,40}. This was further supported by our computational modelling results: the feature hierarchy in DNNs captures increasingly abstract and discriminative features, from edges to textures and whole objects and their spatial arrangements, which all play into the global structure of the scene. We were able to explain up to 60% of variance in realness judgements from just high-level features (related to objects and their configurations,*

supplementary Figure 2). Later, generated scenes which seemed real after initial processing could be more easily discriminated from real scenes based on further recurrent analysis of high-level features and anchor objects (or lack thereof) which informed higher processing areas, in turn influencing downstream predictions and analysis at lower levels."

Minor issues:

Abstract - should be "underlie"

Thank you for noticing, we made corrections accordingly.

The authors refer to the discrimination task from exp. 1 as a "signal detection task", but this is very unspecific as, to my understanding, any detection/discrimination can be called an SDT task. Using this term was quite confusing for me when I was reading the paper for the first time.

We apologize for the confusion. We changed our phrasing to "*two-alternative forced choice task (2AFC)*". We hope that this phrasing is more clear in describing the task used in experiment 1.

Fig. 1 - "Procedures only differed in the task." - not clear what this means, better "differed only in terms of the task performed by participants, but all parameters related to stimuli presentation were the same", or sth similar

We thank the reviewer for the suggestion which indeed improves clarity. We adapted the caption for Figure 1 accordingly.

"there was a significant interaction between presentation duration and true condition" - "true condition" is not clear in this context

Again, we apologize for the confusion here. We now changed each occurrence to "true image condition" (p.6, line 35, p. 15 line 14, 16, & 18). We hope that it is more clear now, that this refers to the ground truth of the image condition (real vs. generated).

Reviewer #2:

The authors investigated if humans can detect generated images from real images at short and long presentation times. They found generated scenes appeared realistic to participants at first glance but were easily discriminated from real scenes at longer presentation durations. They further showed that discriminating between real and generated images seems to be mostly a high level process.

Were the same images used in experiment 1 and 2? Were there any individuals who attended both experiments? If the images were the same, was the order of experiment 1 and 2 counter balanced across shared participants?

We thank the reviewer for these important questions, clarifications and comments. We used the same stimulus set in experiment 1 and experiment 2. Participants that participated in experiment 1 **were not allowed** to participate in experiment 2 (*Participants were unfamiliar with the stimulus material and could only participate in either Experiment 1 or Experiment 2., p.13, line 7*). As participation was managed via SONA we were able to restrict participation to participant IDs that had not previously participated in experiment 1. We added the following sentence to the *Participants* section in the manuscript to make this more clear:

p.13, line 9: "Therefore, there were no participants that participated in both experiments."

I think the procedure of stimulus selection should become more clear. For example, how would the authors generate an image of a specific category with a GAN and be happy with it? Not all the images generated by a GAN are very realistic, so I imagine multiple attempts have been made and then with some criteria the

generated images were selected. Can the authors explain their approach?

We understand that the description of stimulus generation was not clear enough. We purposefully did not perform any selection procedure after randomly generating images from the latent space of each GAN as to get a representative sample that contains the full range of “realism” represented in GAN generated images. We added the following passage to our *Stimuli and design* section in the manuscript to clarify the generation procedure:

p.13, line 19: *“Images were generated by randomly sampling from the latent spaces of the pretrained GANs. Code to generate the same set of images we used in this study can be found via our Open Science Forum (OSF) repository (see Data Availability section). We did not perform any further selection after generating from the random sample. Therefore, we did not remove or replace any of the sampled images, even if they contained artefacts.”*

As you can see, we added the stimulus generation notebook to the OSF repository, where anyone can generate and save to file the exact same set of stimuli by using the provided random state.

What the authors think about the relation of presentation time, neural feedforward/feedback processing and detection of realistic images. Previous work has shown that briefly presented images and backward masking largely disrupt recurrent/feedback processing. I think this can be discussed in the discussion.

Thank you for the comment, in its previous state the manuscript indeed lacked discussion of the involvement of feedforward/feedback processing. We added the following section to the discussion:

p.11, line 12: *“People are able to grasp a scene’s gist (e.g., its basic level category, affordances, and global properties such as navigability), after a few milliseconds^{1,19,20,36}. This fast extraction of meaning relies on both the feed-forward processing of global scene statistics (e.g., statistical spatial layout information)^{19,20} as well as the identification of objects and object constellations in the scene^{10,16,18}. Both processes are assumed to interact with and constrain each other to support analysis at multiple processing levels^{12,37}. [...] After short presentation times of 50 ms, observers were not able to tell apart generated from real scenes. Here, anchor objects – large, stationary objects that are predictive of the location and identity of smaller surrounding objects – contributed to making a scene “feel” like a real scene. Unlike diagnostic objects – which can also be quite small (e.g., toothbrush in bathroom) – anchor objects tend to take up a larger proportion of the scene¹⁶ and therefore contribute to its spatial layout (e.g., a cabinet in the kitchen). We argue that anchor objects inherently influence the statistical spatial layout information of a scene (without needing to be recognized) due to their size and structural properties^{20,38} which in turn provide the basis for scrutinizing a scene’s authenticity during swift feed-forward processing. We can assume that in the 50 ms presentation time condition backward masking largely prohibited recurrent processing and identification of individual objects in our already ambiguous scenes^{39,40}.”*

One interesting analysis would be to train a DNN to discriminate between real and fake scene images, and then use interpretability techniques to show what features were used by the DNN for this discrimination task (similar to Exp. 1).

Interestingly, this is usually how GANs are trained, where a discriminator is trained separately on both real and fake data and tries to correctly classify real images as real and fake images as fake. The discriminator's loss is calculated based on its performance in distinguishing between real and fake data. Using backpropagation, the discriminator's weights are adjusted to minimize this loss, improving its ability to differentiate between real and fake data. At the same time, the generator’s objective is to produce data that is indistinguishable from real data and so its loss is calculated based on how well it fooled the discriminator. So basically, the experiment the reviewer is suggesting here, is what happens during training of the GANs. As the generated images we used here are the result of this kind of training they represent images which most likely would fool any discriminator trained to discriminate between real and fake images. Instead of investigating what features were used by the DNN for this discrimination task, in this study we were more interested in features that emerge across

a range of common computer vision tasks and how they contribute to *human* discrimination between real and generated images.

2nd May 24

Dear Ms Kallmayer,

Thank you for submitting a revised version of your manuscript titled "Making a scene – using GAN generated scenes to test the role of real-world co-occurrence statistics and hierarchical feature spaces in scene understanding.". After careful consideration and discussion with my colleagues, I am sorry to have to tell you that we do not feel that the Reviewers' comments have been sufficiently addressed to justify sending this revision back to the reviewers.

We take this unusual course of action is taken occasionally in order to avoid unproductive rounds of review that ultimately reduce the chances of the manuscript obtaining a fair and objective evaluation.

In order for us to consider this manuscript further please do your best to fully address the following:

Although we recognize that statistics reporting is overall significantly improved, the manuscript does not present positive evidence for null effects, instead interpreting non-significant findings from NHST as evidence for no effect/no difference. Where the interpretation of the findings depends on the absence of an effect or absence of a difference, we require Bayesian statistics or equivalence tests for this purpose (see also: <https://www.nature.com/commpsychol/submit/submission-guidelines#statistical-guidelines>).

We hope to receive your revised version as soon as possible. If you anticipate a delay of more than a month, please let us know.

Please use the link below to submit a suitably revised manuscript and updated response to referees when they are ready.

[link redacted]

Best regards,

Marike

Marike Schiffer, PhD

Chief Editor

Communications Psychology

Communications Psychology is committed to improving transparency in authorship. As part of our efforts in this direction, we are now requesting that all authors identified as ‘corresponding author’ create and link their Open Researcher and Contributor Identifier (ORCID) with their account on the Manuscript Tracking System prior to acceptance. ORCID helps the scientific community achieve unambiguous attribution of all scholarly contributions. You can create and link your ORCID from the home page of the Manuscript Tracking System by clicking on ‘Modify my Springer Nature account’ and following the instructions in the link below. Please also inform all co-authors that they can add their ORCIDs to their accounts and that they must do so prior to acceptance.

3rd Jul 24

Dear Ms Kallmayer,

Your manuscript titled "Making a scene – using GAN generated scenes to test the role of real-world co-occurrence statistics and hierarchical feature spaces in scene understanding." has now been seen by our reviewers, whose comments appear below. In light of their advice I am delighted to say that we are happy, in principle, to publish a suitably revised version in Communications Psychology under the open access CC BY license (Creative Commons Attribution v4.0 International License).

We therefore invite you to revise your paper one last time to address the remaining editorial requests. At the same time we ask that you edit your manuscript to comply with our format requirements and to maximise the accessibility and therefore the impact of your work.

With regard to Reviewer #2's clarification regarding their earlier request, we ask you to incorporate mention of this point in the limitations section; as these concerns do not fundamentally call into question the contribution of the present experiments, we consider this further empirical work beyond the scope of the present paper.

EDITORIAL REQUESTS:

SUBMISSION INFORMATION:

OPEN ACCESS:

Communications Psychology is a fully open access journal. Articles are made freely accessible on publication under a CC BY license (Creative Commons Attribution 4.0 International License). This license allows maximum dissemination and re-use of open access materials and is preferred by many research funding bodies.

For further information about article processing charges, open access funding, and advice and support from Nature Research, please visit <https://www.nature.com/commspsychol/article-processing-charges>

At acceptance, you will be provided with instructions for completing this CC BY license on behalf of all authors. This grants us the necessary permissions to publish your paper. Additionally, you will be asked to declare that all required third party permissions have been obtained, and to provide billing information in order to pay the article-processing charge (APC).

* **DATA AVAILABILITY:**

[link redacted]

Best regards,

Marike

Marike Schiffer, PhD

Chief Editor

Communications Psychology

REVIEWERS' COMMENTS:

Reviewer #1 (Remarks to the Author):

The authors have addressed all my comments, I am happy to recommend the manuscript for publication.

Reviewer #2 (Remarks to the Author):

I appreciate the authors efforts in addressing my previous comments. However, I believe they misunderstood my last comment about training a DNN model to discriminate between real and fake scene images and they have not addressed it yet. I know how GANs are trained, but "deep fake detection" is also an active research field. The idea is that while the GAN discriminator is not capable of discriminating between real and fake images, a larger model with more capacity can be trained on a data set of real and fake images generated by that generative model to discriminate between real and fake. In this case because people were indeed capable of distinguishing between fake and real images, training such model seems feasible.